# Development of A Safety Climate Scale for Geological Prospecting Projects in China

**DOI:** 10.3390/ijerph16061082

**Published:** 2019-03-26

**Authors:** Xiang Wu, Jingqi Gao, Yuanlong Li, Chunlin Wu

**Affiliations:** 1School of Engineering and Technology, China University of Geosciences (Beijing), Beijing 100083, China; wuxiang@cugb.edu.cn (X.W.); gjq1118@cugb.edu.cn (J.G.); 2102170092@cugb.edu.cn (Y.L.); 2Key Laboratory of Deep Geodrilling Technology, Ministry of Land and Resources, Beijing 100083, China; 3School of Economics and Management, Beihang University, Beijing 100191, China; 4Beijing Key Laboratory of Emergency Support Simulation Technologies for City Operations, Beihang University, Beijing 100191, China

**Keywords:** geological prospecting project, safety climate, factor analysis, scale development

## Abstract

The geological prospecting industry has developed rapidly in China over the past few years. It has made outstanding contributions to the discovery of new mineral resources, new energy sources, and the excavation and utilization of resources. However, geological prospecting projects do not have effective safety management measures at present. Moreover, the geological prospecting project has its own traits and features that differ from other industries, leading to the fact that safety management measures in other industries cannot be used in geological prospecting projects. Therefore, development of an effective safety management measuring tool is urgent and necessary. In recent years, safety climate has drawn great attention from scholars, and research results have been successfully applied in construction, coal mining and other industries. Based on the extensive literature review on safety climate as well as its organizational structure and employees’ individual behavior characteristics, this paper first extracted the factor structure of the safety climate and then developed a safety climate scale for geological prospecting projects. This paper used the methods of exploratory factor analysis and reliability analysis to ensure the developed safety climate scale was valid and reliable. The safety climate scale developed has four dimensions, i.e., project leader’s safety commitment, safety institutions, risk response, and employee’s safety attitude, containing a total of 17 measurable items. This study contributes to the current literature by exploring the factor structure of the safety climate for geological prospecting projects, and further provides a scientific basis for improvements in the geological prospecting industry. Meanwhile, the findings not only provide technical support for investigating and analyzing the safety management levels of the geological prospecting industry, but also contribute to the benchmarking standards among different enterprises and projects.

## 1. Introduction

Geological prospecting projects are those projects with the major task of investigation and study focusing on rocks, strata, minerals, groundwater, landforms in certain geographical areas [1]. It carries on the preliminary investigation to the geological structure and the geological condition, and thus guarantees the execution progress of engineering projects. The working methods, working procedures, and working environment of geological prospecting projects determine the types of accident causes found and the probability of accidents, which are completely different from the relatively fixed mining industry. The geological prospecting industry has the characteristics of strong fluidity and high dispersion. Most geological prospecting projects are field work, and the geographical and natural conditions are not under human control, which means the occupational safety level is usually very low. Owing to various hazardous factors, passive protection, and difficult rescue features, geological prospecting accidents occur frequently. However, the existing scientific research and management methods cannot solve the existing safety problems in the geological prospecting project effectively. In view of this, this study takes the characteristics of geological prospecting project into full consideration, so as to explore effective, economical, and feasible safety management measures to improve the safety level of geological prospecting projects.

As the research of safety theories has deepened and engineering practices have improved, it has been found that humans’ unsafe behavior is the major cause of safety accidents. Thus, the measures to control against unsafe behaviors focus on behavioral safety management and influencing behavior through the safety climate [2,3]. The safety climate is an important research field in the area of organizational factors. It has been universally verified by the literature as a priori indicator of the status of organizational safety, and is one of the decisive factors for the success of safety management [4,5,6].

The cause of geological prospecting accidents can be said to be the performance of poor safety climate awareness of geological prospecting projects. Compared with other types of projects, safety management of geological prospecting projects has its particularity. Various reasons, such as long-time outdoor work and complex working environment, result in inefficient safety management and poor safety climate. This climate leads to an absence of understanding safety among upper management and front-line workers. However, if a good safety climate can be established in geological prospecting projects, then project employees will be able to have a deeper understanding of safety and a positive attitude toward productive safety training. Therefore, improving the safety climate of geological prospecting projects can help to effectively prevent geological prospecting accidents. In recent years, many scholars have studied the safety climate of the construction industry, manufacturing industry, chemical industry, transportation industry, and so on [7,8,9,10,11]. However, there are few studies of the safety climate of geological prospecting projects, and the structure of the safety climate is not conclusive.

Zahoor et al. [12] stated that existing safety climate scales cannot be generalized across countries and regions without cultural adjustments. Therefore, this study aims to develop a reliable and valid scale to measure the safety climate of geological prospecting projects, and thus provide valid guidance for future research regarding geological prospecting accident prevention and safety performance improvement. Using the developed safety climate scale, valid safety management can be implemented that ensures that the safety of geological prospecting projects will be valued highly by all employees. This study also aims to contribute to addressing the deficiencies of the current research on geological prospecting industry by systematically studying the factors that influence the safety climate.

## 2. Literature Review

To measure the safety climate of the geological prospecting project, the definition of safety climate, the dimensions of that climate, and the safety climate scale need to be understood. The definition and the dimensions of the safety climate are the bases of the safety climate research and the safety climate scale is an important method of measuring the safety climate. The earliest published papers on safety climate date back to 1951 [13]. After the Chernobyl nuclear accident, Zohar [8] first proposed the concept of the safety climate and defined it as “the perceptions shared by employees in a work environment with risk”. Subsequently, there was more and more research on the safety climate, and different scholars demonstrated various understandings of it. Because of the continuous refinement and deepening of the theories and research in this area, many definitions of have emerged. From the definitions of the safety climate in different periods, important distinctions between the investigations and the research results on safety climate in diverse industries have been established; however, the studies share a common view that the safety climate is a psychological phenomenon. It is closely related to the work environment and the safety status of the enterprise, and it functions as a perception of individuals and organizations on the safety status of their own enterprises within a specific period of time. In this study, based on pervious literature, the safety climate is defined asa mental representation, which is often closely related to the work environment and safety state within an enterprise, and is expressed as a common perception of the safety state of the individual and the organization in the period of time.

Studying the dimensions of the safety climate is an effective way to understand the nature of the safety climate. The research on the safety climate in diverse industries shows differences. Zohar [8] analyzed the safety climate of 20 manufacturing enterprises and then generated eight dimensions said to describe it. Flin et al. [14] studied the maritime safety climate and developed a safety climate measurement scale, which contained 100 items with 19 dimensions. In the aviation industry, Díaz and Cabrera [15] developed a six-dimensional safety climate survey tool that contained 45 items. Dedobbeleer and Beland [16] constructed a safety climate measure for construction sites and concluded that the safety climate has two dimensions: management’s commitment to safety and worker’s involvement in safety. Zhou et al. [17] found four stable safety climate factors in the Chinese construction industry using two surveys.

In the previous literature, it is found that the diverse dimensions of the safety climate in different industries and regions lead to differences in safety climate scales. Therefore, some of the classical safety climate scales are compared and summarized here. Zohar [8] developed a safety climate scale with eight dimensions and 40 items for industrial organizations in Israel. The UK Health and Safety Executive [18] developed a safety climate measurement scale that included a total of 71 items and ten dimensions for the oil and gas industry. Zhou et al. [17] developed a scale for the construction industry with 24 items in 2011. Wang et al. [19] developed a questionnaire for the safety climate in the coal mining industry which contained eight dimensions. In 2015, Wu et al. [20] also developed a scale with 18 items pertaining to construction. Li et al. [21] developed a questionnaire on the safety climate within China’s construction teams, which contained six dimensions and 23 items.

Based on the literature review, the factors structure and the measurement results of safety climate vary in different industries and the geological prospecting projects have their particularity, result in the safety climate scale of the geological prospecting projects is different from other industries to a certain extent. Therefore, this study designed a questionnaire to confirm the most common dimensions of the safety climate and to develop a safety climate scale for geological prospecting projects.

## 3. Methodology

Generally, methodologies such as documentary reviews, questionnaire surveys, interviews, and report measurements are widely utilized in the existing literature. Ordinarily, by using a questionnaire survey, the factor structure or dimensions of the safety climate can be investigated through random sampling online and offline, and the reliability and validity of the safety climate scale can be ensured through interviews. Therefore, this study analyzed the safety climate scale proposed in previous studies through a review of the literature. Based on analysis and summary of the previous literature, combined with interviews with front-line personnel and project leaders, the rationality of the safety climate structure was guaranteed and a questionnaire survey on the structural dimensions of the safety climate was analyzed. To explore the various dimensions that affect the safety climate of the geological prospecting project and to design the geological prospecting project safety climate measurement scale, the exploratory factor analysis (EFA) method and the Statistical Product and Service Solutions (SPSS, International Business Machines Corporation, Armonk, NY, USA) software tool were used to analyze the data acquired from the questionnaire survey. In the first stage, EFA was applied to analyze the initial survey data to obtain the structural dimensions of the safety climate and to improve the preliminary safety climate scale. In the second stage, SPSS was used to analyze the responses to confirm the reliability of the improved scale and, finally, to generate the safety climate factor structure and the formal safety climate scale.

### 3.1. Theory Construction

Based on the existing research on safety climate, combined with the characteristics of the geological prospecting projects, this study identified 11 dimensions that affect the safety climate of the geological prospecting project, namely manager’s safety commitment, manager’s safety behavior, safety attitude, safety regulation, safety communication, safety training, risk response, safety participation, work environment, safety institution, and workers’ behavior.

The two safety climate dimensions that were included from previous research were the manager’s safety commitment and the manager’s behavior. According to the industry characteristics of the geological prospecting project, most projects regard the manager as the project leader, so this study changed these two dimensions from “manager’s safety commitment” and “manager’s safety behavior” to “project leader’s safety commitment” and “project leader’s safe behavior”, respectively.

The dimensions are outlined in detail as follows:(1)*Project leader’s safety commitment*. The managers’ safety commitment as a dimension of the safety climate is generally recognized in the literature. Pilbeam et al. [22] found that the project leaders’ safety commitment can affect employees’ compliance with safety. Other research works [20,21,23] has also pointed out that the manager’s safety commitment has an impact on the safety climate. Therefore, this study chose project leader’s safety commitment as a dimension of the geological prospecting project safety climate and integrated the actual situation of the industry into the design of the safety climate scale.(2)*Project leader’s safety behavior*. The project leader’s safety behavior mainly refers to the project leader’s own safety behavior, as well as the supervision and monitoring of the employees’ behaviors and the concerns over employees’ safety. In previous studies, Neal et al. [24] measured the safety climate through employee-perceived managers’ concern for employees’ safety. In this study, the project leader’s safety behavior was further examined as it is a major dimension of the safety climate of a geological prospecting project.(3)*Safety attitude*. Safety attitude can also be understood as an individual’s emphasis on safety. Safety attitudes have been widely recognized as part of the safety culture [20]. Where safety technologies cannot reduce accident rates even further, research on safety attitudes can provide further precautionary measures to reduce accidents to an acceptable level [25]. At present, research on safety attitudes has been making continuous progress in transportation and in the coal-mining industry. However, there are few studies on safety attitudes in geological prospecting projects. As for the effects of safety attitudes in this safety climate, this study could only study them in accord with the previous literature.(4)*Safety regulations*. Safety regulations refer to the basic rules established by enterprises for all types of workers according to the nature of production and the characteristics of technical equipment; these are also the main basis of safety education for workers. Guo et al. [26] found that safety regulation in the construction industry is the basis for identifying risks. In addition, many researchers have found that the development and improvement of safety regulation will affect the enterprise’s safety climate. Lu et al. [27] and Zhou et al. [17] both recognized the influence of safety regulations on safety climate in their research.(5)*Safety communications*. Safety communications between front-line workers and project leaders as well as between staff and managers help to correct unsafe behaviors and create a good safety climate. Fugas et al. [2] found that the safety communication among team members or between project leaders and team members enhances the shared perception of each project member regarding the safety climate of the project. Glendon and Litherland [28] also showed that safety communication has a considerable impact on the safety climate. Mutual reminders among team members are helpful in changing unsafe behaviors, and these are an important manifestation of the safety climate of geological prospecting projects.(6)*Safety training*. A huge safety risk is usually related to a lack of safety knowledge, and safety training can both regulate employees’ attitudes toward safety and supplement safety knowledge [29]. Many studies have also realized this point, i.e., safety training is regarded as a key issue in the study of safety climates [2,20]. In combination with the characteristics of the geological prospecting project, it is necessary to study the safety climate from the perspective of safety training.(7)*Risk response*. Geological prospecting projects focus on improving the ability to identify potential hazards and pay closer attention to safety protection skills and escape methods. Some studies consider risk response a dimension that affects the safety climate [30]. Others regard risk cognition and emergency response as two important dimensions of the safety climate [31]. In this study, risk response was chosen as one dimension of the safety climate.(8)*Safety participation*. Safety participation is one dimension of safety behavior, which can evaluate the safety performance of an enterprise, and it is also a direction for further study of the safety climate. Cooper et al. [28] and Wu et al. [30,32] considered workers’ safety participation to have a great impact on the safety climate and regarded it as a key dimension of safety climate. Safety participation can be regarded as a deeper level of safety behavior outside of the employee’s job role. It refers to employees’ active participation in safety activities for the benefit of the organization, such as the experienced employees helping new employees understand safety precautions, employees actively reflecting on safety issues, suggestions to superiors, and so on, in addition to adopting routine work safety requirements.(9)*Work environment*. The safety climate is strongly linked to workers’ commitment to safety and health in the work environment [24]. Ji et al. [33] studied the safety climate of the construction industry, where they regarded the environment as a major element from the point of view of the research. Owing to the changing and uncertain work environment of geological prospecting projects, most sites are inhospitable, remote, and harsh. As a result, it is very difficult to have perfect safety protection for project members during their working hours. As for the safety accidents caused by environmental change, the result is usually not satisfactory even if the management has made considerable safety investments. Therefore, when considering the safety climate of the geological prospecting project, this study cannot ignore its highly variable working environment.(10)*Safety institutions*. Health and safety management requires the establishment of a health and safety institution sector and safety professionals to perform related tasks. Zohar [8] mentioned the impact of safety institutions on the organization’s safety climate in his initial study of the safety climate. His research proved that the status of safety committees and the safety manager’s position in the organization has a major impact on the safety climate. Wu et al. [32] also emphasized the important role of safety institutions in accident prevention. This study on the safety climate of geological exploration projects takes the characteristics of safety institutions into account.(11)*Workers’ behavior*. In an enterprise, the behavior of workers can influence both employees themselves and the safety climate. By studying the safety behaviors of employees, this study analyzed their impact on organizations’ safety climates. In addition, Li et al. [21] measured the workers’ behavior in their study and approved its role in improving the safety climate of the construction teams. In this study, the impact of workers’ behavior was recognized as a dimension of the safety climate.

### 3.2. Preliminary Development of Scale

To achieve empirical results, this study used a questionnaire. The questionnaire items rated safety climate on a five-point Likert scale ranging from 1 = “strongly disagree” to 5 = “strongly agree.” There were three questionnaire items in the measurement scale that needed to be converted to rank order, namely, “strongly disagree” recorded as 5 points, and “strongly agree” recorded as 1 point; these helped to screen out incomplete questionnaires and reinforced reliability and validity. The higher the scores, the more likely they were to reflect a better safety climate in the geological prospecting project. In the initial design of the safety climate scale, 11 of the most common dimensions were explored, and the questionnaire was developed based on these dimensions.

Following the previous literature, the initial scale had 35 items overall. Except for the project leader’s safety commitment and the risk response, which had four items each, other dimensions had only three items each, and they were not the final dimensions. It is only with the EFA and the SPSS measurements of reliability and validity that the final safety climate scale could be determined.

### 3.3. Research Samples and Procedures

Employees in the geological prospecting project were interviewed and an online survey was subsequently conducted after exploring and validating the safety climate dimensions and measurement items of the geological prospecting project. The final questionnaire survey retained 35 items.

The questionnaire survey was separated into two steps. First, to understand the project better, the initial questionnaire survey used a large sample of relatively decentralized geological prospecting practitioners to conduct the empirical research and send out the questionnaires to their staff. It was stated at the beginning of the questionnaire that this questionnaire will only be used for research purposes. The questionnaires were repeated both online and offline. The EFA was applied to analyze the initial survey data to improve the original safety climate scale. Subsequently, this study used SPSS to confirm the reliability of the improved scale and, finally, to obtain the formal safety climate scale.

In this study, a total of 115 questionnaires were sent out in the first step. The questionnaire was deleted if the time taken to fill out the questionnaire was too long or too short, the answers were the same, or the questionnaire was incomplete. Moreover, answers to three items which the rank order are converted, were checked. A response was deleted if the answers to all the three items were found in contract with most of the other responses. There were 105 valid questionnaires retained, for a response rate of 91.3%. Among them, males accounted for the majority, 83.8%, of valid respondents. The types of respondents include workers, contractors’ frontline managers, and project senior leaders. The types of respondents include workers, contractors’ frontline managers, and project senior leaders. For these respondents, 67.6% of the total were aged 30 or below. With regard to educational background, 83.8% had a high school education. In addition, 74.3% had no accident experience. All the criteria in the following tests were based on the work of Wu [34], Hair et al. [35] and Schreiber et al. [36], unless otherwise specified. The significance level was set at 0.05.

Then, SPSS was used, and the four dimensions of the safety climate scale were finally determined: project leader’s safety commitment, safety institutions, risk response, and employees’ safety attitude. The formal safety climate scale consists of four dimensions and 17 items.

## 4. Results

### 4.1. Exploratory Factor Analysis

The main task of exploratory factor analysis is to extract and synthesize the information overlap of the original variables into a factor. First, the degree of discrimination of each item was analyzed by using item analysis. Reverse scoring items on the scale, and then finding the total scores was undertaken so that the scale reflects the same tendencies. The results were ranked from high to low. The first 27% were chosen as the highest scoring group, and the last 27% were chosen as the lowest scoring group. An independent sample t-test was used to analyze the empirical data, with the resulting t value being the critical ratio (CR). In this measurement, t-test values of a7, a11, a27, and a33 were all less than 3, and items with *t* < 3 (*p* < 0.05) were omitted. A total of 32 items met the requirements.

Factor analysis requires a strong correlation between the original variables; otherwise, if the original variables are independent, the degree of correlation is very low, there is no overlap of information, there can be no common factor, and there is no need for factor analysis. Thus the Kaiser–Meyer–Olkin (KMO) coefficient and Bartlett’s test were used to analyze whether there was a correlation between the original variables, that is, to determine whether they were suitable for factor analysis. The results showed that the KMO coefficient was 0.906 and the chi square test was 2546.211 (df = 496, *p* < 0.001), indicating that the data could be subjected to factor analysis.

We analyzed 32 items and then determined public factors based on eigenvalues and scree plot (Figure 1). The results showed that six factors could be obtained; the eigenvalues were 13.062, 2.425, 1.497, 1.234, 1.084 and 0.986, respectively, accounting for 71.579% of the variance in total. Cox and Cox [37] stated that the standard loading of factors should not be less than 0.4. The requirements of this study are more conservative. The principle is that if commonality is more than 0.2, the factor loading is more than 0.5.

In exploratory factor analysis, the first common factor includes the two dimensions obtained above: the project leader’s safety commitment, and the project leader’s safety behavior. As the problem is biased towards the project leader’s project planning this study called this dimension “project leader’s safety commitment.” In addition, the item in workers’ behavior (WB) “Every team member plays an important role in safety production” is summarized in the same dimension as the item in the work environment dimension, namely, the “work environment” dimension. The working environment includes both the natural environment, as well as the human environment in which employees communicate. In addition, the items of safety participation and safety attitude consist of the fourth dimension of thee factor analysis, and their classification can be referred to as “employees’ safety attitude.” According to the above principles, the study merged one dimension and finally retained five dimensions: project leader’s safety commitment (PLC), safety institutions (SI), risk response (RR), employees’ safety attitude (ESA), and work environment (WE) respectively. The cumulative explanation rate is 71.327%. The loading factors, communality, the factor eigenvalues, and the cumulative explanatory variables of each variable are shown in Table 1 and Table 2.

Through analysis of the principal component factor loading, the correlation between items a30, a8, a29 and the total scale was found to be low. The factor loading on item a35 was also found to be lower than 0.5. However, considering that the factor loading on item a35 meets the Cox’s standard and it has a certain value, it was retained temporarily. Moreover, the correlation of the five dimensions was tested. The person coefficient of the working environment on the total scale is only 0.625, compared with the project leader’s safety commitment, safety institutions, risk response, and employees’ safety attitude; which total scale is low, but the relevant level is acceptable, so this dimension can be temporarily retained.

### 4.2. Reliability Analysis

In this study, the validity of the safety climate measurement scale was analyzed by SPSS 22.0 and the internal consistency of safety climate scale was verified by Cronbach’s α. The test results are shown in Table 3.

In general, Cronbach’s α should be around 0.7 or higher. In the reliability analysis, the fifth factor’s Cronbach’s α is 0.385. According to Wu’s research, when Cronbach’s α is less than 0.5, the reliability of this dimension is not satisfactory, so the dimension should be omitted [34]. After omitting this factor, Cronbach’s α of the total scale is 0.930, which shows that the scale has good reliability in measuring the safety climate of the geological prospecting project and can be used. After the reliability analysis of the scale, the four dimensions and 17 items of the formal scale were finally retained; the questionnaire items in the safety climate scale of the geological prospecting project are in Appendix A.

## 5. Discussion

### 5.1. Summary of Major Findings

The main objective of this study was to develop a safety climate scale for geological prospecting projects. After conducting the EFA and reliability analysis, the final scale consisted of four dimensions, i.e., project leader’s safety behavior, safety institutions, risk response, and employee’s safety attitude, with a total of 17 items. This study examined the construct validity of the scale by the application of the EFA and also verified the content validity by using reliability analysis. Therefore, the final scale was an effective tool to measure the geological prospecting projects’ safety climate.

The dimensions of the geological prospecting project safety climate are quite different from other industries. This may be due to the complex technology of the geological prospecting project, thus employees need to have a high job skills and knowledge reserves to be competent in prospecting work. As geological prospecting project employees generally have higher education than average, the knowledge level of employees and their outlook on life, worldview, and values make their compliance with and understanding of the safety rules more established.

The influence of safety training for employees who have a stronger than usual sense of safety is limited. Coworkers generally pay more attention to safety and have formed a good safety climate, so the role of coworkers’ behavior is not apparent. In contrast, project leader’s safety commitment, safety institutions, risk response, and employee’s safety attitude have a significant impact on the safety climate of the geological exploration project. The leadership’s impact has been studied and proved as a significant factor in safety management by many scholars in different fields, such as construction projects. Project leaders’ rewards and punishments with regard to safety behaviors will encourage safety production by the front-line staff and enhance the safety climate level of the project. Meanwhile, the importance attached by the manager to safety institutions has a significant impact on the safety climate of the enterprise. Secondly, the attention of safety institutions is conducive to ensuring the safety supervision of project safety managers so as to improve the safety climate in the operation. Third, the dimension of risk response is in line with the characteristics of the industries that have strong liquidity in geological prospecting projects. The effective emergency measures for accidents have a huge impact on reducing accident losses, thus affecting the safety climate of geological prospecting projects. In the progress of a geological prospecting project, with the more reasonable and effective risk emergency measures, there will gradually exist a more positive impact on the safety climate of geological prospecting projects. Another factor that most affects the safety climate of geological prospecting projects is the employee’s safety attitude, which is extremely important. If employees take safety seriously and respond to safety policies, a project’s safety climate will be greatly improved.

### 5.2. Theoretical Implications

The safety climate plays an important role in promoting the safety management system and reducing unsafe behavior. Therefore, exploring the dimensions of the safety climate and measuring it accurately can effectively improve the safety climate in geological prospecting projects. This study has contributed to the theoretical implications of the following two aspects:This study contributes to the current occupational safety research by developing a valid and reliable factor structure of the safety climate of geological prospecting projects. It is different from pervious safety climate scale and more suitable for geological prospecting industry. By providing a uniform measuring criterion, this scale will facilitate future safety climate research in the industry. By re-examining the validity and reliability of the safety climate scale of geological prospecting industry with a larger sample of employed people in a future study, the research will promote the safety climate and safety management in the geological prospecting industry.The safety climate scale developed contains four dimensions: project leader’s safety behavior, safety institutions, risk response, and employee’s safety attitude. There are 17 items in total, and the scale demonstrates high validity and reliability (see Appendix A, Table A1). The dimensions showed that the major factors of the safety climate proved that the importance of the role of project leaders in geological prospecting projects’ safety management was the same as in other industries.

### 5.3. Practical Implications

Moreover, this study also gives insights into safety management practices within the geological prospecting industry. The practical implications include the following two aspects:The effective safety climate scale derived from this study is a comprehensive tool suited for geological prospecting projects. It can be used to examine the safety perception of project leaders and employees. The measurement scale serves as an important tool for safety climate benchmarking among different geological prospecting enterprises, and thus can boost overall safety in the industry as a whole. Moreover, with reasonable modification, the safety climate scale could be used in other industries to help to improve safety performance.The implementation of the safety climate scale would provide project leaders and employees with rich feedback to realize the pros and cons in safety policies in the companies. According to the findings of the questionnaire survey, it is also recommended that the safety training and education of geological prospecting employees should be strengthened, their work pressure should be reduced, and project leader’s commitment to safety should be increased. These are the most crucial measures to promote the safety climate of the geological prospecting projects currently.

### 5.4. Limitations and Future Research Directions

Geological prospecting is an enormous industry, which includes water, engineering, mineral resources, and oil and gas. All of them are involved in the safety management of geological prospecting projects. Future research should focus on studies in these different areas as well as on different aspects of the safety climate.

Zhang et al. [38] have pointed out that two types of safety climate scale are mainly used in China at present: one directly refers to the relatively mature measurement scales used abroad, and the other is the scale compiled according to the characteristics of domestic enterprises. This study compiled a safety climate scale according to the characteristics of geological prospecting projects, but Zhang et al. [38] have indicated that the structure and content of such scales are determined by previous perceptual knowledge and experience, which are influenced by the subjective decisions of relevant experts or authors; this affects their scientific nature. As the safety climate scale of geological prospecting project is seldom studied, whether the scientific character of the developed scale is subject to the authors’ subjective influence needs further study.

## 6. Conclusions

In this study, the safety climate factor structure and measuring scale of geological prospecting projects were developed initially. Before this study, a safety climate measurement tool that suited the geological prospecting projects had not been introduced. Through the stepwise screening of factor analysis and validity analysis, four safety climate dimensions of the geological prospecting project were identified, namely project leader’s safety commitment, safety institutions, risk response, and employee’s safety attitude. Moreover, with regard to the safety climate measurement scale that this study developed, validity reaches up to 0.930 and reliability is also quite high, indicating that the scale is robust and can provide support for the analysis of the project’s safety management status.

The practical significance of this study is to providing an effective tool to guide safety management in geological prospecting projects. By measuring this climate, the results can be used as a reference frame to guide normative safety behaviors and effective safety management. Furthermore, the development of the safety climate scale will contribute to future studies on the safety climate and safety management not only in the industry itself but also in the other industries which have similar characteristics.

## Figures and Tables

**Figure 1 ijerph-16-01082-f001:**
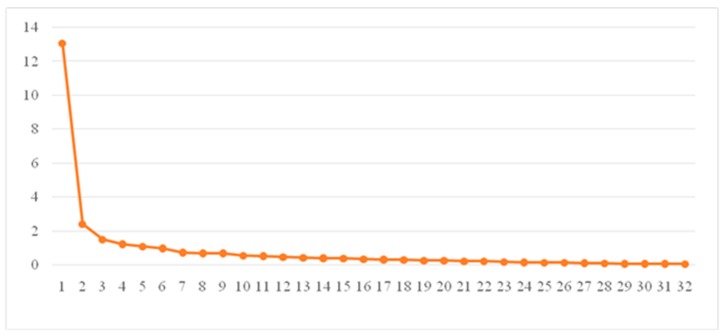
Scree plot.

**Table 1 ijerph-16-01082-t001:** Principal component and factor loadings.

Items	Factor Loading						Communality
	PLC	SI	RR	ESA	WE	WB	
a5	0.824						0.716
a6	0.625						0.727
a2	0.708						0.595
a3	0.635						0.651
a12	0.632						0.738
a1	0.625						0.562
a32		0.790					0.742
a33		0.733					0.712
a35		0.468					0.607
a31		0.533					0.610
a23			0.843				0.697
a24			0.662				0.669
a22			0.522				0.817
a25				0.724			0.754
a10				0.586			0.488
a26				0.501			0.816
a15				0.557			0.846
a30					1.188		0.431
a8					0.498		0.468
a29						−1.076	0.397
Eigenvalue	13.062	2.425	1.497	1.234	1.084	0.986	
Cumulative % of explanatory variance	46.082	54.639	59.920	64.275	68.099	71.579	

Note: PLC-project leader’s safety commitment; SI-safety institutions; RR-risk response; ESA-employees’ safety attitude; WE-work environment; WB-workers’ behavior.

**Table 2 ijerph-16-01082-t002:** The result of the correlation analysis.

	PLC	SI	RR	ESA	WE
**PLC**	1				
**SI**	0.590 **	1			
**RR**	0.661 **	0.620 **	1		
**ESA**	0.671 **	0.647 **	0.745 **	1	
**WE**	0.466 **	0.541 **	0.481 **	0.521 **	1
**Scale**	0.847 **	0.827 **	0.818 **	0.867 **	0.625 **

Note: ** indicate *p* < 0.01. PLC-project leader’s safety commitment; SI-safety institutions; RR-risk response; ESA-employees’ safety attitude; WE-work environment; WB-workers’ behavior.

**Table 3 ijerph-16-01082-t003:** The Cronbach’s α of safety climate scale.

Factor	Cronbach’s α	Number of Items
**PLC**	0.873	6
**SI**	0.823	4
**RR**	0.863	3
**ESA**	0.860	4
**WE**	0.385	3
**Scale**	0.930	20

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
