# Peer review of "Development of A Safety Climate Scale for Geological Prospecting Projects in China"

_ijerph, 2019, doi:10.3390/ijerph16061082_

Round 1

Reviewer 1 Report

It is a well-drafted manuscript; however, it has many downsides.

The introduction and literature review sections are poor and not explicative. The necessity of this research should have been elaborated more. Surprisingly, the concept of geological prospecting, around which this study revolves, is not explained in detail.

Likewise, the research method is not explained well. Before conducting the main survey, no pilot study is carried out with safety experts to validate the items of the questionnaire. How the quality of results from the questionnaire was ascertained? How were anonymity and confidentiality ensured? It is not mentioned how the treatment of the collected Data was carried out. 

Principle component analysis is not performed well. The number of factors should have been selected based on the Scree plot, Horn’s parallel analysis, or Eigenvalues.

Please see the following paper for developing a safety climate questionnaire and how to carry our exploratory factor analysis.

Zahoor, H., Chan, A. P., Utama, W. P., Gao, R., & Memon, S. A. (2017). Determinants of safety climate for building projects: SEM-based cross-validation study. Journal of construction engineering and management143(6), 05017005.

Also explain how this research adds something significant to previous research in the field of safety climate.

Author Response

The authors would like to thank the reviewer for the constructive comments on our manuscript “Development of A Safety Climate Scale for Geological Prospecting Projects in China”. The manuscript has been carefully revised following the comments. Please find the detailed responses below.

 (In the following text, the comments of reviewers are shown in blue and italic font to be distinguished from the authorsresponses.)

Responses to Reviewer #1

Reviewer 1

1-1 It is a well-drafted manuscript; however, it has many downsides.

The introduction and literature review sections are poor and not explicative. The necessity of this research should have been elaborated more. Surprisingly, the concept of geological prospecting, around which this study revolves, is not explained in detail.

Response: Thank you for your comments. Within the introduction and literature review sections in this revised manuscript, we have added the concept and the specified characteristics of geological prospecting projects, in order to explain the necessity of this research more clearly. The revised description is as follows:

Lines 40-49, Page 2:

“Geological prospecting projects are those projects with the major task of investigation and study focusing on rocks, strata, minerals, groundwater, landforms in certain geographical areas [1]. It carries on the preliminary investigation to the geological structure and the geological condition, and thus guarantees the execution progress of engineering projects. The working methods, working procedures, and working environment of geological prospecting projects determine the types of accident causes found and the probability of accidents, which are completely different from the relatively fixed mining industry. The geological prospecting industry has the characteristics of strong fluidity and high dispersion. Most geological prospecting projects are field work, and the geographical and natural conditions are not under human control, which means the occupational safety level is usually very low.”

Please find similar revisions when we explain the necessity of this study in Lines 63-66, Page 2, Lines 75-76, Page 2, Lines 99-102, Page 3 and Lines 124-127, Page 3.

1-2 Likewise, the research method is not explained well. Before conducting the main survey, no pilot study is carried out with safety experts to validate the items of the questionnaire. How the quality of results from the questionnaire was ascertained? How were anonymity and confidentiality ensured? It is not mentioned how the treatment of the collected Data was carried out.

Response: Thank you for this constructive advice. We apologize for unclear description. We have interviewed the employees in the geological prospecting projects to ensure the availability of the questionnaire before online survey. The final issued questionnaire retained 35 items (Lines 254-257, Page 6).

We have added some details as follows, in order to make the research method more specific, and addressed your above concerns and comments: 

Line 261-262, Page 6:

“The beginning of the questionnaires mentioned that this questionnaire is only used for research purposes and would guarantee the anonymity of the participants.”

Line 266-270, Page 6:

“In this study, a total of 115 questionnaires were sent out in the first step. The questionnaire was deleted if the time taken to fill out the questionnaire was too long or too short, the answers were the same, or the questionnaire was incomplete. Moreover, answers to three items which the rank order are converted, were checked. A response was deleted if the answers to all the three items were found in contract with most of other responses.”

1-3 Principle component analysis is not performed well. The number of factors should have been selected based on the Scree plot, Horn’s parallel analysis, or Eigenvalues.

Response: We apologize for unclear description. We have added the scree plot as Figure 1 (Page 7) to show the obtained six factors. The sixth point in the scree plot in the inflection point and the eigenvalues were 13.062, 2.425, 1.497, 1.234, 1.084 and 0.986, respectively, accounting for 71.579% of the variance in total.

1-4 Please see the following paper for developing a safety climate questionnaire and how to carry our exploratory factor analysis.

Zahoor, H., Chan, A. P., Utama, W. P., Gao, R., & Memon, S. A. (2017). Determinants of safety climate for building projects: SEM-based cross-validation study. Journal of construction engineering and management, 143(6), 05017005.

Response: Thank you for your advice. We have quoted this reference in introduction section and have modified the results section in this revised manuscript.

Lines 75-76, Page 2:

“Zahoor et al. [12] stated that existing safety climate scales cannot be generalized across countries and regions without cultural adjustments.”

Lines 324-331, Page 9:

Through analysis of the principal component factor loading, the correlation between items a30, a8, a29 and the total scale was found to be low. The factor loading on item a35 was also found to be lower than 0.5. However, considering that the factor loading on item a35 meets the Cox’s standard and it has a certain value, it was retained temporarily. Moreover, the correlation of the five dimensions was tested. The person coefficient of the working environment on the total scale is only 0.625, compared with the project leader’s safety commitment, safety institutions, risk response, and employees’ safety attitude; which total scale is low, but the relevant level is acceptable, so this dimension can be temporarily retained.

1-5 Also explain how this research adds something significant to previous research in the field of safety climate.
Response: Based on the literature review, we have concluded the 11 most common dimensions of the safety climate. We have also developed the final safety climate scale of the geological prospecting projects, which contributes to the diversity of the study on safety climate. The study on the dimensions of safety climate in the geological prospecting projects can provide reference to future study on other industries’ safety climate.

We revised Section 5.2 to explain the implications of this research.

Lines 381-389, Page 10:

The safety climate plays an important role in promoting the safety management system and reducing unsafe behavior. Therefore, exploring the dimensions of the safety climate and measuring it accurately can effectively improve the safety climate in geological prospecting projects. This study has contributed to the theoretical implications of the following two aspects.

This study contributes to the current occupational safety research by developing a valid and reliable factor structure of the safety climate of geological prospecting projects. It is different from pervious safety climate scale and more suitable for geological prospecting industry. By providing a uniform measuring criterion, this scale will facilitate future safety climate research in the industry. By re-examining the validity and reliability of the safety climate scale of geological prospecting industry with a larger sample of employed people in a future study, the research will promote the safety climate and safety management in the geological prospecting industry.

The safety climate scale developed contains four dimensions: project leader’s safety behavior, safety institutions, risk response, and employee’s safety attitude. There are 17 items in total, and the scale demonstrates high validity and reliability (see Appendix A). The dimensions showed that the major factors of the safety climate proved that the importance of the role of project leaders in geological prospecting projects’ safety management was the same as in other industries.

Reviewer 2 Report

Dear author(s), 

Thank you for the opportunity to review your work on safety climate. As a safety researcher, I was excited to read your scale development. I believe this paper has much promise, but just needs a bit of clarification and editing in a few areas. I detail my comments below: 

I am not convinced yet that the safety climate of geological prospecting projects requires a new safety climate scale. I am aware that certain industries have developed their own safety climate scales, but I would like you to show why this context also needs a new scale. Based on your final scale, the only dimension that is context specific is risk response. Additionally, this scale was created based on the adaptation of other safety climate scales, which furthers my point that other scales could work in this context. You need to show that the use of other safety climate scales are inadequate for this context.

It is unclear how you went from the 11 dimensions that affect safety (which you found in the literature) to your two dimensions. Need more of a rationale for this decision. 

I was also confused why you explained each of the 11 dimensions but said that you only studied two of them. This section needs more clarity. 

How is safety participation different than safety behaviors? These seem almost synonymous (see page 5). 

Also, you need to site which scales you are adapting from the literature in your scale development. Where did you get these 35 items from?

Need to provide more detail on how you went about the preliminary development of your scale. This part was vague. 

I was confused on how your interviews informed your study. Also, please re-write the second paragraph under 3.3 so that it is clearer. What was the second stage? What happened after the first stage? Why did you make these choices?

Fix typo on page 7, line 290

On your tables, write the dimensions instead of the abbreviations.

Need to talk about what you did with the low factor loading on item a35

Again in your results, you need to show that this safety climate scale is different, right now it looks like you retested other scholars' scales. 

What do you mean by "the more perfect the emergency measures are with regard to risk, the better the safety climate of the project" (p. 9, line 348)

I wish you the best of luck with this manuscript and hope to see it in print sometime. 

Author Response

Author Responses to Reviewer’ Comments

The authors would like to thank the reviewer for the constructive comments on our manuscript “Development of A Safety Climate Scale for Geological Prospecting Projects in China”. The manuscript has been carefully revised following the comments. Please find the detailed responses below.

 (In the following text, the comments of reviewers are shown in blue and italic font to be distinguished from the authors responses.)

Responses to Reviewer #2

Reviewer 2

Dear author(s),

Thank you for the opportunity to review your work on safety climate. As a safety researcher, I was excited to read your scale development. I believe this paper has much promise, but just needs a bit of clarification and editing in a few areas. I detail my comments below:

I am not convinced yet that the safety climate of geological prospecting projects requires a new safety climate scale. I am aware that certain industries have developed their own safety climate scales, but I would like you to show why this context also needs a new scale. Based on your final scale, the only dimension that is context specific is risk response. Additionally, this scale was created based on the adaptation of other safety climate scales, which furthers my point that other scales could work in this context. You need to show that the use of other safety climate scales are inadequate for this context.

Response: Thank you for taking your time to read our manuscript. We have carefully considered your comments and we have added several sentences to explain the necessity of a new safety climate scale is needed in geological prospecting projects as follows:

Line 46-49, Page 2:

“The geological prospecting industry has the characteristics of strong fluidity and high dispersion. Most geological prospecting projects are field work, and the geographical and natural conditions are not under human control which means the occupational safety level is usually very low.”

Line 63-66, Page 2:

“Compared with other industries, safety management of geological prospecting project has its significant particularity. Various reasons, such as long-time outdoor work and complex working environment, result in an inefficient safety management and a poor safety climate.”

Although other certain industries have developed their own safety climate scale, the geological prospecting industry has its own features, the differences between its working methods, working procedures, working environment, accident causes and others industries make this research has its necessityLines 43-46, Page 2; Lines 124-129, Page 3.

Because of geological prospecting industry particularity, other safety climate scales are not adequate. Therefore, we have developed a new safety climate scale for geological prospecting projects in this paper.

2-1 It is unclear how you went from the 11 dimensions that affect safety (which you found in the literature) to your two dimensions. Need more of a rationale for this decision.

I was also confused why you explained each of the 11 dimensions but said that you only studied two of them. This section needs more clarity.

Response: We apologize for unclear description. We have concluded 11 dimensions based on the existing research. Due to the characteristics of the geological prospecting projects, the project leaders are regarded as the safety managers, and thus we have changed the names of 2 dimensions, from manager’s safety commitment to project leader’s safety commitment and from manager’s safety behavior to project leader’s safe behavior. The rest nine dimensions, namely safety attitude, safety regulation, safety communication, safety training, risk response, safety participation, work environment, safety institution, and workers’ behavior, do not need to be changed.

Line 154-159, Page 4:

“The two safety climate dimensions that were included from previous research were the manager’s safety commitment and the manager’s behavior. According to the industry characteristics of the geological prospecting project, most projects regard the manager as the project leader, so this study changed these two dimensions from “manager’s safety commitment” and “manager’s safety behavior” to “project leader’s safety commitment” and “project leader’s safe behavior”, respectively.”

2-2 How is safety participation different than safety behaviors? These seem almost synonymous (see page 5).

Response: We apologize for unclear description. Safety participation is one dimension of safety behavior. Safety behavior can be divided into two dimensions, namely safety compliance and safety participation. Safety participation emphasizes the subjective initiative of safety behavior. We have revised the description of the safety participation as follows:

Lines 207-215, Page 5:

Safety participation. Safety participation is one dimension of safety behavior, which can evaluate the safety performance of an enterprise, and it is also a direction for further study of the safety climate. Cooper et al. [28] and Wu et al. [29,31] considered workers’ safety participation to have a great impact on the safety climate and regarded it as a key dimension of safety climate. Safety participation can be regarded as a deeper level of safety behavior outside of the employee’s job role. It refers to employees’ active participation in safety activities for the benefit of the organization, such as the experienced employees helping new employees understand safety precautions, employees actively reflecting on safety issues, suggestions to superiors, and so on, in addition to adopting routine work safety requirements.

2-3 Also, you need to site which scales you are adapting from the literature in your scale development. Where did you get these 35 items from? Need to provide more detail on how you went about the preliminary development of your scale. This part was vague.

Response: We apologize for unclear description. We have got these 35 items based on the 11 dimensions which we have summarized form pervious literature as well as the characteristics of geological prospecting projects. In the development process of the geological prospecting project safety climate scale, we have combined the characteristics of the geological prospecting projects and modified the safety climate scales’ items, especially in the language expression. The 35 items in the initial scale are more aligned with the geological prospecting projects’ features and also easier to understand for the employees, and thus the reliability of the safety climate scale can be guaranteed. Though it is difficult to correspond the 35 items with original literature, the 11 dimensions of our initial questionnaire are supported by references. The original references are as follows.

2.         Fugas, C.S.; Silva, S.A.; Melia, J.L. Another look at safety climate and safety behavior: Deepening the cognitive and social mediator mechanisms. Accident Analysis and Prevention 2012, 45, 468-477, doi:10.1016/j.aap.2011.08.013.

8.         Zohar, D. Safety climate in industrial organizations: theoretical and applied implications. The Journal of applied psychology 1980, 65, 96-102, doi:10.1037/0021-9010.65.1.96.

17.       Zhou, Q.A.; Fang, D.P.; Mohamed, S. Safety Climate Improvement: Case Study in a Chinese Construction Company. J. Constr. Eng. Manage.-ASCE 2011, 137, 86-95, doi:10.1061/(asce)co.1943-7862.0000241.

Please find other original references in the References list from No.20 to No.33.

2-4 I was confused on how your interviews informed your study. Also, please re-write the second paragraph under 3.3 so that it is clearer. What was the second stage? What happened after the first stage? Why did you make these choices?

Response: First, we have revised the description of the first paragraph under 3.3. The purpose of the interviews is to ensure the availability of the questionnaire. The communication with the employees helped us to modify or delete inappropriate items.

Lines 254-257, Page 6:

“Employees in the geological prospecting project were interviewed and an online survey was subsequently conducted after exploring and validating the safety climate dimensions and measurement items of the geological prospecting project. The final questionnaire survey retained 35 items.”

Second, we have made below modifications to make the research procedures clearer.

Line 258-265, Page 6:

“The questionnaire survey was separated into two steps. First, to understand the project better, the initial questionnaire survey used a large sample of relatively decentralized geological prospecting practitioners to conduct the empirical research and send out the questionnaires to their staff. The beginning of the questionnaires mentioned that this questionnaire is only used for research purposes and would guarantee the anonymity of the participants. The questionnaires were repeated both online and offline. The EFA was applied to analyze the initial survey data to improve the original safety climate scale. Subsequently, this study used SPSS to confirm the reliability of the improved scale and, finally, to obtain the formal safety climate scale.”

Please find other modifications in Lines 276-278, Page 7.

2-5 Fix typo on page 7, line 290

On your tables, write the dimensions instead of the abbreviations.

Response: Thank you for your revision and your advice. We have tried to modify the tables, but found that it would mess up our tables’ layout and make they difficult to read. Therefore, we have added the notes for abbreviations of dimensions below the tables (Line 319-320, Page8 ; Line 322-323, Page 9) and kept the abbreviations in our tables.

2-6 Need to talk about what you did with the low factor loading on item a35

Response: Thank you for your comments. We have added the details of how we did on item a35.

Line 325-327, Page 9:

“The factor loading on item a35 was also found to be lower than 0.5. However, considering that the factor loading on item a35 meets the Cox’s standard and it has a certain value, it was retained temporarily.”

2-7 Again in your results, you need to show that this safety climate scale is different, right now it looks like you retested other scholars' scales.

Response: Thank you for your comments. In the exploratory factor analysis, we have integrated the items and the dimensions into five factors (Line 303-314, Page 7). We made those choices based on the full consideration of the particularity of the geological prospecting industry. Though our initial climate scale was developed based on the previous literature and safety climate scales of other industries, the final safety climate scale we have obtained is more suitable for geological prospecting projects. The safety climate scales are various in different industries, even in the same kind of industries in diverse countries and regions. Therefore, the final safety climate scale in our paper referred to, but differed from, the scales of previous research works.

Lines 385-392, Page 10:

“This study contributes to the current occupational safety research by developing a valid and reliable factor structure of the safety climate of geological prospecting projects. It is different from pervious safety climate scale and more suitable for geological prospecting industry. By providing a uniform measuring criterion, this scale will facilitate future safety climate research in the industry. By re-examining the validity and reliability of the safety climate scale of geological prospecting industry with a larger sample of employed people in a future study, the research will promote the safety climate and safety management in the geological prospecting industry.”

2-8 What do you mean by "the more perfect the emergency measures are with regard to risk, the better the safety climate of the project" (p. 9, line 348)

Response: We apologize for unclear description. We have modified the description.

Line 374-376, Page 10:

“In the progress of a geological prospecting project, with the more reasonable and effective risk emergency measures, there will gradually exist a more positive impact on the safety climate of geological prospecting projects.”

2-9 I wish you the best of luck with this manuscript and hope to see it in print sometime.

Response: Thank you for your best wishes, and again thanks for taking time to read our manuscript.

Reviewer 3 Report

The manuscript is stimulating, appropriate topic, informative and developed a safety climate scale for geological prospecting projects. It will be of great interest to many people. The investigators have provided evidence of rigorous methods, significant results and acceptable conclusions. The article is well structured and the ideas are presented in the clear and sufficient way. The most valuable part of the research is the questionnaire in the methodology and determined the dimension of safety climate. The article is also strictly connected with the scope of the journal and therefore it is worth of publishing in the present form.

I have only two comments:

1. Page 11, line 407 – “This study‘s practical significance lies in” is not appropriate. I suggest that” The practical significance of this study is to”

2. I have some doubts about the universality of the chosen items for the countries with different work culture of geological prospecting project. The final scale can be implemented in China, but could it be effective in other countries or regions?

Author Response

Author Responses to Reviewer’ Comments

The authors would like to thank the reviewer for the constructive comments on our manuscript “Development of A Safety Climate Scale for Geological Prospecting Projects in China”. The manuscript has been carefully revised following the comments. Please find the detailed responses below.

 (In the following text, the comments of reviewers are shown in blue and italic font to be distinguished from the authors responses.)

Responses to Reviewer #3

Reviewer 3

3-1 Page 11, line 407 – “This study‘s practical significance lies in” is not appropriate. I suggest that” The practical significance of this study is to”

Response: Thank you and sorry for our carelessness. We have revised this expression and we have also checked the whole manuscript to delete any possible typos or unclearness.

Lines 439-440, Page 11:

“The practical significance of this study is to providing an effective tool to guide safety management in geological prospecting projects.”

3-2 I have some doubts about the universality of the chosen items for the countries with different work culture of geological prospecting project. The final scale can be implemented in China, but could it be effective in other countries or regions?

Response: Thank you for your comments. In this study, the safety climate scale is designed for geological prospecting projects in China, and thus it reflects some typical characteristics of its specific setting. We have explained in the title that this is for China's geological prospecting industry. Considering working environment and cultural background, the dimensions and indicators of safety climate may not be all the same across different countries and industries, but they still have significant similarities in different settings. Thus, this study can also give references for safety attitude research in different countries and industries.

Round 2

Reviewer 1 Report

Thank you for addressing the highlighted observations in the manuscript entitled "Development of A Safety Climate Scale for Geological Prospecting Projects in China".

The manuscript is now in good form.

However, authors are requested to explain how their developed safety climate scale is different from already developed scales. Likewise, the outcome of the interviews is not lined with the results of the literature review. Thirty-five safety climate items, selected from the literature review, are not enumerated. Type of respondents depending upon various stakeholders, such as contractors, clients and consultants/designer, is not elaborated.

Importantly, authors are requested to do proofreading of paper. Also, fix the typo and remove grammatical errors. Few are highlighted below:

Line-100: Change "asa" to "as a".

Line-268: Change the sentence, as: It was stated at the beginning of the questionnaire that this questionnaire will only be used for research purposes.....

Line-278: ....found in contrast with most of the other responses....

Author Response

The authors would like to thank the reviewer for the constructive comments on our manuscript “Development of A Safety Climate Scale for Geological Prospecting Projects in China”. The manuscript has been carefully revised following the comments. Please find the detailed responses below.

In particular, the revisions in the revised manuscript are highlighted.

(In the following text, the comments of reviewers are shown in blue and italic font to be distinguished from the authors responses.)

Responses to Reviewer #1

Reviewer 1

Thank you for addressing the highlighted observations in the manuscript entitled "Development of A Safety Climate Scale for Geological Prospecting Projects in China". The manuscript is now in good form.

However, authors are requested to explain how their developed safety climate scale is different from already developed scales. Likewise, the outcome of the interviews is not lined with the results of the literature review. Thirty-five safety climate items, selected from the literature review, are not enumerated. Type of respondents depending upon various stakeholders, such as contractors, clients and consultants/designer, is not elaborated. Importantly, authors are requested to do proofreading of paper.

Response: Thank you for your comments and we apologize for our unclear description. We want to address your above four concerns one by one as follows.

Firstly, although our initial climate scale was developed based on the previous literature and safety climate scales of other industries, the final safety climate scale we have obtained is more suitable for geological prospecting projects. The safety climate scales are various in different industries, even in the same industry among different countries and regions. Therefore, the final safety climate scale in our paper referred to, but differed from, the scales of previous research works. We have elaborated this issue in more details in Line 386-393, Page 10 of the revised manuscript.

Secondly, based on the previous literature, the initial scale had 35 items. After the interviews, the final safety climate questionnaire retained 35 items (Line 248-257, Page 6). This indicates that the outcome of the interviews is in line with the results of the literature review.

Third, the 35 items are not enumerated because we thought it only provides unnecessary details. These 35 items are based on the 11 dimensions we have summarized form pervious literature, and also incorporate the characteristics of geological prospecting projects.

The details of the 35 items are as follows. The project leader’s safety commitment has four items, which focus on the importance attached by the project leader. The project leader’s safety behavior has three items. This dimension is to test the specific safety management actions of the project leader. The three items in safety attitude mainly include the workers’ understanding of accidents and the relationship between safety and production efficiency. The safety regulations and safety training dimensions both have three items, which pay attention to both project leader’s and workers’ attitude towards the safety regulations and training. The three safety communication items were mainly set up based on the communication of safety issues among employees and the communication between project leader and workers. Risk response has four items, including risk prevention, accident response, and cognition and emotion. Safety participation has three items, focusing on employees’ role in the promotion of safety climate, while workers’ behavior focusing on other workers’ attitude and behavior in the workplace. The three items of the work environment dimension were developed according to the safety evaluation of the outdoor workplace. And the items of safety institutions focus on the position and the functions of safety institution in the geological prospecting projects (Lines 160-238, Pages 4-6).

Last but not least, the respondents were mainly from the sites and management teams of geological prospecting projects. The types of respondents include workers, contractors’ frontline managers, and project senior leaders. We have added the details in this revised manuscript (Line 272, Page 6).

Also, fix the typo and remove grammatical errors. Few are highlighted below:

Line-100: Change "asa" to "as a".

Line-268: Change the sentence, as: It was stated at the beginning of the questionnaire that this questionnaire will only be used for research purposes.....

Line-278: ....found in contrast with most of the other responses....

Response: We have fixed the typos and removed grammatical errors. We have rechecked this manuscript thoroughly to prevent any similar problems. Thank you very much for your help!

Reviewer 2 Report

Dear authors, 

Thank you for your changes to the manuscript. I believe the revised manuscript is clearer and makes a stronger contribution to the extant literature. I look forward to seeing this piece in print. 

Author Response

We thank this reviewer very much for your continuous help and support to our paper.